# RNA-FM: Flow-Matching Generative Model for Genome-wide RNA-Seq Prediction

Yaxuan Song [1]   Jianan Fan [1]   Tianyi Wang [1]   Qiuyue Hu [1]   Hang Chang [2 3]   Heng Huang [4]   Weidong Cai [1]

## Abstract

Histopathology whole-slide images (WSIs) are routinely acquired in clinical practice and contain rich tissue morphology but lack direct molecular architecture and functional programs defining pathological states, whereas RNA sequencing (RNA-seq) provides genome-wide transcriptional profiles at substantial cost, thereby motivating WSI-based genome-wide transcriptomic prediction. Existing approaches for predicting gene expression from WSIs predominantly rely on deterministic regression with one-to-one mapping, limiting their ability to capture biological heterogeneity and predictive uncertainty. We propose RNA-FM, a flow-matching generative framework for genome-wide bulk RNA-seq prediction from WSIs. RNA-FM formulates transcriptomic prediction as a continuous-time conditional transport problem, learning a velocity field that maps a simple prior to the target gene expression distribution conditioned on morphologies. By integrating pathway-level structure, RNA-FM enables scalable and biologically interpretable genome-wide gene expression imputation. Extensive experiments demonstrate that RNA-FM consistently outperforms state-of-the-art approaches while maintaining biological meaningfulness. Code is available at https://github.com/YXSong000/RNA-FM.

## 1. Introduction

RNA sequencing (RNA-seq) has become one of the most widely used genome-wide transcriptomic profiling tech-

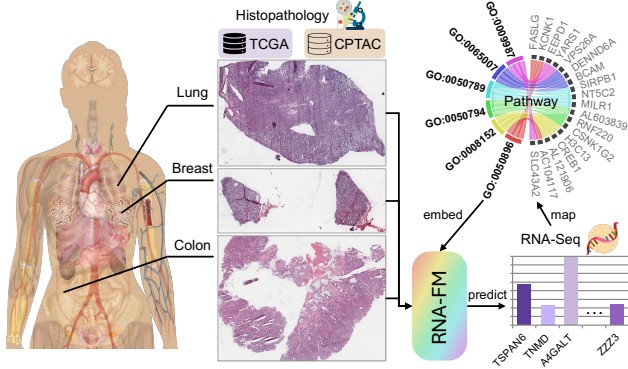

*Figure 1.* RNA-FM predicts genome-wide bulk RNA-seq profiles from histopathology whole-slide images across multiple anatomical sites using a flow-matching generative model. By incorporating pathway-level representations, RNA-FM enables scalable and biologically interpretable transcriptomic inference conditioned on histopathological morphology.

niques in modern biomedical research and clinical diagnostics. By capturing global transcriptional activity from tissue samples, RNA-seq supports a broad range of applications, including disease classification, biomarker discovery, pathway analysis, and treatment decision-making (Li & Wang, 2021; Wang et al., 2026; 2025a). Despite its utility, RNA-seq remains costly and resource-intensive, limiting its routine use at scale in clinical practice. Meanwhile, whole-slide images (WSIs) are routinely acquired in clinical practice and provide high-resolution representations of tissue morphology (Pizurica et al., 2024; Fan et al., 2025), yet they do not directly encode underlying molecular states or genetic functional programs (Greten et al., 2024; Chen et al., 2021). This complementary relationship motivates growing interest in computationally bridging histology and transcriptomics: by learning to predict gene expression directly from WSIs, it becomes possible to enrich routine slide-based diagnosis with genome-wide molecular insights *in silico*.

Several recent studies have explored this direction by leveraging deep learning models to regress bulk gene expression profiles from WSIs (Schmauch et al., 2020; Alsaafin et al., 2023; Pizurica et al., 2024). While these approaches show promising imputation quality, they predominantly formulate the task as a deterministic regression problem, producing point estimates for each gene. As a result, they neglect

---

[1]School of Computer Science, The University of Sydney, Australia [2]Biological Systems and Engineering Division, Lawrence Berkeley National Lab, USA [3]Berkeley Biomedical Data Science Center, Lawrence Berkeley National Lab, USA [4]Department of Computer Science, University of Maryland College Park, USA. Correspondence to: Jianan Fan <jianan.fan@sydney.edu.au>, Weidong Cai <tom.cai@sydney.edu.au>.

*Proceedings of the 43$^{rd}$ International Conference on Machine Learning*, Seoul, South Korea. PMLR 306, 2026. Copyright 2026 by the author(s).

predictive uncertainty and suppress biologically meaningful variability in gene expression (Zhu et al., 2025). In practice, WSIs from many anatomical regions, e.g., breast, lung, and colon, exhibit substantial heterogeneity arising from variations in cell-type composition, tissue architecture, and tumor microenvironmental context (Chan et al., 2023). A deterministic one-to-one mapping from image to gene expression often fails to capture both inter-patient and intra-tumoral heterogeneity (Pizurica et al., 2024), ultimately limiting imputation performance. This challenge is further amplified in bulk RNA-seq, where aggregation across heterogeneous cell populations increases ambiguity in the histology-to-transcriptomics mapping.

These limitations motivate a probabilistic generative formulation that explicitly models the conditional distribution of gene expression given histopathological context, rather than collapsing predictions to a single point estimate. However, genome-wide RNA-seq introduces an additional scalability challenge: gene expression profiles span over 20,000 genes, making it computationally infeasible to treat each gene as an independent token or embedding at the genome scale. To address both dimensionality and interpretability, pathway-aware structure leverages prior biological knowledge by aggregating genes into functionally coherent groups. Structuring models around known biological pathways not only reduces the dimensionality but also ensures that the genes are organized in interpretable and biologically meaningful units during imputation, as illustrated in Figure 1.

In this work, we propose **RNA-FM**, a pathway-aware **F**low-**M**atching generative framework for bulk **RNA**-seq prediction from whole-slide histopathology images. RNA-FM formulates transcriptomic prediction as a continuous-time conditional transport problem, learning a velocity field that maps a simple prior distribution to the target bulk RNA-seq distribution conditioned on morphology-derived features. By adopting flow matching, RNA-FM avoids iterative denoising and likelihood estimation, enabling stable and efficient training in high-dimensional gene expression space while naturally supporting uncertainty-aware generation.

Our contributions are summarized as follows:

- We propose a novel algorithm, RNA-FM, to predict genome-wide bulk RNA-seq profiles from histopathology images via a pathway-aware flow-matching generative model.
- RNA-FM is able to capture both inter-patient and intra-tumoral heterogeneity, enabling uncertainty-aware and one-to-many mappings between tissue morphology and gene expression of bulk-level RNA-seq.
- We introduce pathway-aware representations that patchify >20,000 genes into biologically interpretable gene-set tokens, enabling scalable genome-wide prediction without compromising model robustness.

- Extensive experiments across various anatomical regions, pathway-level analysis, and external validation cohorts demonstrate that RNA-FM consistently outperforms state-of-the-art deterministic regression methods in both prediction accuracy and generalization.

## 2. Related Work

### 2.1. Bulk-Level RNA-Seq Prediction from WSIs

Recent advances in computational pathology (Song et al., 2023; Fan et al., 2025; Zhao et al., 2025) have explored predicting bulk RNA-seq profiles directly from WSIs. Early work HE2RNA (Schmauch et al., 2020) introduces the first large-scale framework to predict genome-wide gene expression directly from H&E-stained WSIs using weakly supervised learning. By aggregating tile-level features into slide-level representations, HE2RNA achieved significant gene-wise correlations across multiple cancer types, establishing the feasibility of transcriptome-wide prediction from histology images. Subsequent studies improved feature aggregation using attention mechanisms. tRNAsformer (Alsaafin et al., 2023) proposed a transformer-based multiple-instance learning model that captures contextual relationships among image patches for bulk RNA-seq prediction. This approach achieved improved convergence and competitive predictive performance while enabling downstream tasks such as image-based search and classification. In a more recent study, the SEQUOIA framework employed linearized attention to efficiently aggregate foundation-model-derived histological features for RNA-seq prediction (Pizurica et al., 2024), representing a significant improvement in deterministic WSI-based transcriptomic inference.

### 2.2. Generative Models for Transcriptomic Prediction

Diffusion-based generative models, including denoising diffusion probabilistic models (DDPMs) and score-based methods, model data generation as a denoising process that reverses a predefined noise diffusion (Ho et al., 2020; Song et al., 2021). These models have achieved strong performance in image synthesis (Wang et al., 2025c) and have recently been applied to biological domains such as single-cell RNA-seq (Luo et al., 2024), spatial transcriptomics (Zhu et al., 2025; Song et al., 2026), where their ability to capture uncertainty and multimodality is particularly valuable. However, standard diffusion models present practical limitations for genome-wide bulk-level RNA-seq prediction from histopathology images. Training and inference require iterative denoising across many noise levels, leading to high computational cost for genome-wide (over 20,000) expression vectors (Ho et al., 2020; Kim et al., 2025a). Moreover, score-based objectives can be difficult to estimate accurately in high-dimensional, highly correlated transcriptomic data (Hyvärinen & Dayan, 2005; Song et al., 2021), and

the hundreds of sampling steps during inference limit scalability (Nichol & Dhariwal, 2021). These challenges are exacerbated by the aggregated bulk RNA-seq profiles. Flow-matching generative models provide an efficient alternative by directly learning a continuous-time velocity field that transports samples from a simple prior to the target distribution (Lipman et al., 2023; Ma et al., 2024; Wang et al., 2025b). By matching velocities instead of denoising noisy samples, flow matching avoids explicit likelihood estimation and iterative sampling, resulting in improved training stability and faster inference via ordinary differential equation solvers. These properties make flow-matching particularly suitable for genome-wide transcriptomic generation.

### 2.3. Gene Ontology and Pathway

Biological pathways encode curated functional relationships among genes and serve as a basis for transcriptomic interpretation. Classical methods such as Gene Set Enrichment Analysis (GSEA) (Subramanian et al., 2005) and single-sample GSEA (ssGSEA) leverage pathway annotations from Gene Ontology (Ashburner et al., 2000; Consortium, 2025) and Reactome (Joshi-Tope et al., 2005) to interpret expression profiles in terms of biological processes (Subramanian et al., 2005; Barbie et al., 2009). These pathway-level representations are widely used for molecular subtyping, disease characterization, and clinical decision-making (Song et al., 2024). In deep learning, pathway knowledge has been integrated in architectures (Kim et al., 2025b), including pathway-level embeddings (Jaume et al., 2024), graph neural networks (Ma & Wang, 2024), and attention-based pathway aggregation (Zhang et al., 2022). Such methods improve generalization by constraining models to respect known biological organization, enabling coherent information propagation across related biological functions.

## 3. Methodology

In this study, we address the task of predicting genome-wide bulk RNA-seq profiles from histopathology WSIs using a probabilistic generative framework to enhance the biologically meaningful variability in gene expression. Specifically, given a histopathology image $y$ and its corresponding bulk RNA-seq gene expression profile $x$, our objective is to model the conditional distribution $p(x \mid y)$, capturing the inherent biological heterogeneity beyond deterministic point estimates. To this end, we propose RNA-FM, a pathway-aware flow-matching generative model that learns a time-dependent velocity field that transports a prior distribution to the target bulk RNA-seq distribution, conditioned on histopathology image features. By incorporating pathway-level structure, RNA-FM enables scalable and biologically interpretable genome-wide modeling. Figure 2 illustrates the overall framework of RNA-FM.

### 3.1. Histopathology Image Feature Extraction

Due to their extremely high spatial resolution, WSIs are rarely encoded directly by a feature extractor. To address this challenge, we adopt a multiple-instance learning (MIL) paradigm to construct compact, informative slide-level representations. Specifically, each WSI with a magnification of $\times 20$ ($0.5 \mu m$ per pixel) is partitioned into non-overlapping tiles of size $256 \times 256$ pixels. The Otsu threshold approach (Otsu et al., 1975) is utilized to filter out tiles with a white background (Pizurica et al., 2024). Up to $4000$ tiles are randomly selected by discarding those with more than 20% background or low contrast. Tile-level feature representations are extracted by a feature extractor $\Phi$, with $d$ dimensions. In this work, we utilize either a ResNet-50 backbone pretrained on ImageNet or the UNI (Chen et al., 2024) model pretrained on over 100 million pathology images. The compact slide-level representations are obtained by $k$-means clustering tile-level features within the slide into $k = 100$ clusters. The slide-level representation is the aggregation of 100 clusters with morphologically similar tiles, $y \in \mathbb{R}^{100 \times d}$.

### 3.2. Pathway-Aware Representation for Gene Profile

Since directly modeling genome-wide RNA-seq gene expressions as independent tokens is computationally infeasible and biologically suboptimal. To address this, RNA-FM incorporates pathway-level patchify representations curated from a Gene Ontology (GO) functional set and explores inter-pathway interactions to facilitate coherent information propagation across related biological functions.

**Gene-to-Pathway Embedding.** Let $x \in \mathbb{R}^G$ denote a bulk RNA-seq profile with $G$ genes, where $G = 20820$ for the genome-wide RNA-seq in this task. We define $P$ pathways as index sets of genes $\{\mathcal{P}_i\}_{i=1}^{P}$, where $\mathcal{P}_i \subset \{1, \ldots, G\}$, and a background set $\mathcal{B} \subset \{1, \ldots, G\}$ containing genes not covered by any GO knowledgebase. For each pathway $i$, the gene set $x_{\mathcal{P}_i} \in \mathbb{R}^{|\mathcal{P}_i|}$ is computed as a pathway token $p_i$ with $h$ dimensions of the pathway latent space, by a two-layer MLP $f_i : \mathbb{R}^{|\mathcal{P}_i|} \to \mathbb{R}^h$:

$$p_i = f_i(x_{\mathcal{P}_i}) + e_i, \quad i = 1, \ldots, P. \tag{1}$$

Stacking all pathway tokens yields the pathway token matrix:

$$P(x) = [p_1; \ldots; p_P] \in \mathbb{R}^{P \times h}. \tag{2}$$

Similarly, the background genes $x_{\mathcal{B}}$ are embedded using an MLP $f_{\mathrm{bg}} : \mathbb{R}^{|\mathcal{B}|} \to \mathbb{R}^h$ and a learnable bias $e_{\mathrm{bg}} \in \mathbb{R}^h$:

$$b = f_{\mathrm{bg}}(x_{\mathcal{B}}) + e_{\mathrm{bg}}, \quad b \in \mathbb{R}^h. \tag{3}$$

**Inter-pathway Graph Encoding.** To model dependencies between pathways (functional gene sets), we define a

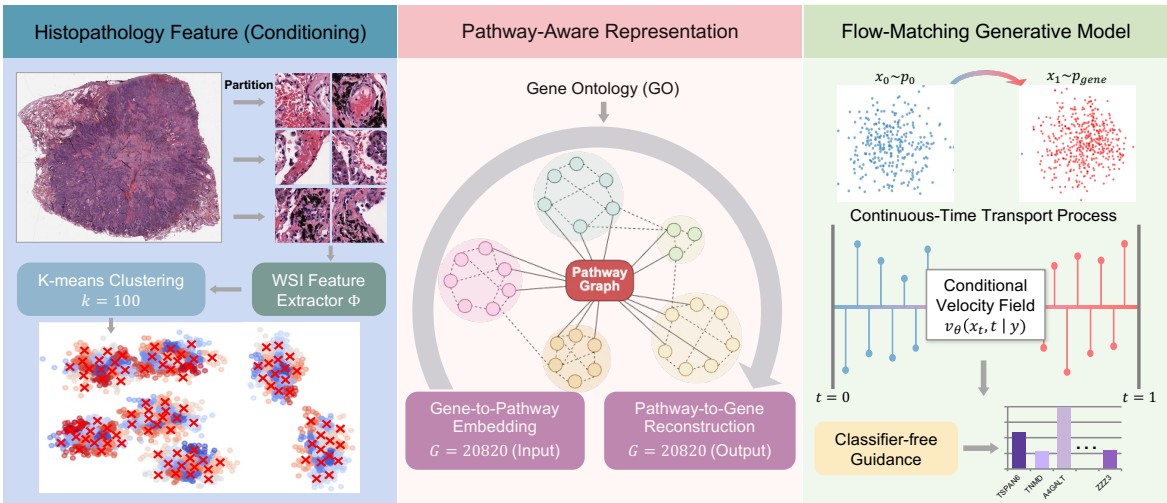

*Figure 2.* The proposed RNA-FM framework for genome-wide bulk RNA-seq prediction from histopathology images. WSIs are partitioned and encoded into slide-level features via clustering, which conditions a flow-matching generative model. Genes are grouped into GO biological processes and modeled through a pathway graph, enabling structured gene-to-pathway embedding and reconstruction. RNA-FM learns a continuous-time conditional transport from a simple prior to the target RNA-seq distribution, with classifier-free guidance applied during sampling for improved conditional fidelity.

pathway graph with an adjacency matrix $A \in \{0,1\}^{P \times P}$ encoding which pathways are allowed to interact, where $A_{ij} = 1$ indicates that pathway $i$ containing overlapped gene(s) to pathway $j$. To obtain a stable and symmetric graph representation, $A$ is adjusted by a series of operations of symmetrization, self-looping, and normalization, obtaining $A_{\text{norm}}$ as the result. Subsequently, an additive attention mask $M \in \mathbb{R}^{P \times P}$ is constructed following rules:

$$M_{ij} = \begin{cases} 0, & (A_{\text{norm}})_{ij} > 0, \\ -\infty, & (A_{\text{norm}})_{ij} = 0. \end{cases} \quad (4)$$

Inter-pathway dependencies are established via graph multi-head self-attention, and the mask $M$ is applied to the attention logits so that disallowed pathway pairs receive zero attention probability:

$$P(x)' = \text{MultiHeadAtten}(P(x), P(x), P(x); M), \quad (5)$$

where $M$ enforces pathway-level graph constraints by preventing attention between unconnected pathways, ensuring that information exchange follows known biological functional relationships. Finally, the block applies residual connections with a learnable residual scale $\alpha$ and a feed-forward network (FFN):

$$P(x)' = P(x) + \alpha \cdot \text{Drop}(P(x)'), \quad (6)$$

$$P(x)'_{\text{out}} = P(x)' + \text{FFN}(\text{LayerNorm}(P(x)')). \quad (7)$$

In our implementation, FFN is a two-layer MLP with GELU activation.

**Pathway-to-Gene Reconstruction.** The pathway-specific gene prediction $\hat{x}_{\mathcal{P}_i} \in \mathbb{R}^{|\mathcal{P}_i|}$ is computed by the pathway $i$

head $h_i(\cdot)$, and the background gene prediction $\hat{x}_{\mathcal{B}} \in \mathbb{R}^{|\mathcal{B}|}$ is computed by the background head $h_{bg}(\cdot)$. We assemble the prediction $\hat{x} \in \mathbb{R}^G$ by adding with overlap averaging:

$$\hat{x}_g = \frac{\sum_{i:\, g \in \mathcal{P}_i} \hat{x}_{\mathcal{P}_i, \, \text{idx}_i(g)} \; + \; \mathbb{I}[g \in \mathcal{B}] \, \hat{x}_{\mathcal{B}, \, \text{idx}_{bg}(g)}}{\sum_{i:\, g \in \mathcal{P}_i} 1 \; + \; \mathbb{I}[g \in \mathcal{B}]}, \quad (8)$$

where $\text{idx}_i(g)$ maps gene index $g$ to its position within the ordered index set $\mathcal{P}_i$ (and analogously for $\text{idx}_{bg}$).

### 3.3. RNA-Seq Flow-Matching Generative Modeling

RNA-FM implements genome-wide bulk RNA-seq generation via the conditional probability flow model as a time-dependent transport process. Given a bulk RNA-seq expression vector over $G$ genes $x \in \mathbb{R}^G$ and the corresponding histopathology image $y$, we consider a time-indexed, in the range of 0 to 1, random variable $x_t$ evolving according to the ordinary differential equation (ODE):

$$\frac{dx_t}{dt} = v_\theta(x_t, t \mid y), \quad t \in [0, 1], \quad (9)$$

where $v_\theta$ denotes the pathway-aware graph network that maps the current gene expression state, diffusion time, and histopathology-derived features to a velocity field governing the generative flow.

Following the general flow-matching formulation (Lipman et al., 2023; Ma et al., 2024), RNA-FM defines a probability path $\{x_t\}_{t \in [0,1]}$ that connects a prior distribution (standard multivariate Gaussian) $p_0 = \mathcal{N}(0, I_G)$ to the conditional gene expression distribution $p_{\text{gene}}(x \mid y)$. Specifically, we sample endpoint pairs $(x_0, x_1)$ such that $x_0 \sim p_0$ and $x_1 \sim p_{\text{gene}}(\cdot \mid y)$, and construct an interpolation of the form:

$$x_t = \alpha(t) \, x_1 + \sigma(t) \, x_0, \quad (10)$$

where $\alpha(t)$ and $\sigma(t)$ are smooth scalar functions satisfying appropriate boundary conditions: $\alpha(0) = 0$, $\sigma(0) = 1$ and $\alpha(1) = 1$, $\sigma(1) = 0$. The corresponding target velocity path for RNA-seq expression along this path is given by:

$$v^{\star}(x_t, t) = \frac{\mathrm{d}x_t}{\mathrm{d}t} = \dot{\alpha}(t)\, x_1 + \dot{\sigma}(t)\, x_0, \qquad (11)$$

where $\dot{\alpha}(t)$ and $\dot{\sigma}(t)$ denote time derivatives, and the velocity field $v(x, t \mid y)$ is defined as the conditional expectation:

$$v(x, t \mid y) = \mathbb{E}[\dot{x}_t \mid x_t = x, y],$$
$$= \dot{\alpha}(t)\, \mathbb{E}[x_1 \mid x_t = x, y] \; + \; \dot{\sigma}(t)\, \mathbb{E}[x_0 \mid x_t = x, y]. \qquad (12)$$

The further detailed proof for the formulation Equation (12) is illustrated in Appendix Section A. Therefore, RNA-FM learns the conditional velocity field $v_\theta$ for gene expression prediction by minimizing the flow-matching objective:

$$\mathcal{L}_{\text{RNA-FM}}(\theta) = \mathbb{E}_{x_0, x_1, t}\Big[\big\|v_\theta(x_t, t \mid y) - v^{\star}(x_t, t)\big\|^2\Big], \qquad (13)$$

where $x_0 \sim p_0$, $x_1 \sim p_{\text{gene}}(\cdot \mid y)$, and $t \sim \mathcal{U}[0, 1]$.

During inference, we employ classifier-free guidance (CFG) (Ho & Salimans, 2021) to strengthen conditioning on histopathology features and improve the fidelity of generated gene expression profiles. Specifically, since RNA-FM is trained with conditional dropout on the image representation, jointly learning conditional and unconditional velocity fields is promoted by combining them with a CFG scale $s$:

$$v_\theta^{\text{cfg}}(x_t, t \mid y) = v_\theta(x_t, t \mid \varnothing) + s\Big(v_\theta(x_t, t \mid y) - v_\theta(x_t, t \mid \varnothing)\Big), \qquad (14)$$

where $v_\theta(x_t, t \mid y)$ and $v_\theta(x_t, t \mid \varnothing)$ denote the conditional and unconditional velocity predictions, respectively.

## 4. Experiments and Results

### 4.1. Datasets and Implementation Details

**TCGA.** In this study, we conduct experiments on cancer types (1) LUAD, (2) BRCA, and (3) COAD at lung, breast, and colon regions, respectively, using paraffin-embedded (FFPE) WSIs and corresponding gene expression data retrieved from the public TCGA archive via the Genomic Data Commons (GDC) Data Portal[1] to train and evaluate performance on RNA-FM. Specifically, the three datasets, LUAD, BRCA, and COAD, contain **536**, **1130**, and **455** WSI-gene-paired samples, respectively. Following the work (Pizurica et al., 2024), we partition each dataset into non-overlapped 5 splits and perform 5-fold cross-validation for evaluation; the sample IDs for each split are provided for reproducibility.

**CPTAC.** To evaluate the generalizability of RNA-FM, we leverage unseen WSI samples[2] and corresponding gene

expression[1] data of the same cancer types as TCGA utilized from the Clinical Proteomic Tumor Analysis Consortium (CPTAC) as an external validation cohort. CPTAC contains **336**, **133**, and **105** WSI-gene paired samples for cancer types LUAD, BRCA, and COAD, respectively.

For the histopathology image feature extractor, the dimension of the WSI feature $d$ is either 2048 or 1024 when using ResNet-50 or UNI, respectively. During training, the model backbone consists of 7 DiT blocks with 8 attention heads and a 512-dimensional shared latent space for WSI and transcriptomic data; details are in Appendix Section B. For RNA-seq, we use FPKM-UQ gene expression values, transformed by $x \rightarrow log_2(x + 1)$ to avoid bias toward genes with high expression. The 20,820 genes are classified into protein-coding genes, microRNAs, and long non-coding RNAs; specifically, protein-coding genes account for 85% of all analyzed genes (Pizurica et al., 2024). RNA-FM is trained using the AdamW optimizer with a learning rate $1e^{-4}$ with up to 2000 epochs. The batch size is set to 32. We sample the diffusion time steps uniformly from $[0, 1]$ that $\alpha_t = 1 - t$, $\sigma_t = t$. For gene expression imputation, we use the probability-flow ODE formulation with an Euler solver over 20 steps. The CFG guidance scale $s$ is set to 2 to strengthen conditioning on WSI features. An exponential moving average (EMA) of model parameters is maintained throughout training and used for evaluation. We use Pearson Correlation Coefficient (PCC) and Root Mean Squared Error (RMSE) as evaluation metrics throughout experiments. The calculation of PCC, RMSE, and highly predictive gene selection is illustrated in Appendix Section C. We implement RNA-FM in PyTorch and train/evaluate models on an NVIDIA RTX A6000 (48GB) GPU.

### 4.2. Cross-validation Performance

Table 1 presents cross-validation results comparing RNA-FM with representative baseline methods across TCGA-LUAD, TCGA-BRCA, and TCGA-COAD datasets and multiple subsets of predictive genes. Additional results for predictive gene subsets (T200 and T20) are illustrated in Appendix Section D. RNA-FM consistently demonstrates strong performance: combined with the UNI feature extractor, RNA-FM achieves the highest overall accuracy in all settings, indicating robust generalization across anatomical sites and gene subset sizes. On LUAD, RNA-FM with UNI features achieves the highest PCC and lowest RMSE across all gene subsets, with particularly notable improvements for more highly predictive gene subsets. Similar performance trends are observed on BRCA and COAD, where RNA-FM consistently surpasses prior approaches across all evaluation metrics, including tRNAsformer and SEQUOIA. These significant improvements for reduced gene subsets suggest that RNA-FM effectively captures biological transcriptional signals rather than relying on averaged predictions. Overall,

---

[1]https://portal.gdc.cancer.gov
[2]https://www.cancerimagingarchive.net

*Table 1.* Comparison of cross-validation experiments with baselines across datasets in TCGA. The T1000, T500, T100, and T50 denote the top-1000, 500, 100, and 50 predictive genes, respectively. The **best results** are in bold; the second-best results are underlined.

| Dataset | Method | Feature Extractor | T1000 PCC↑ | T1000 RMSE↓ | T500 PCC↑ | T500 RMSE↓ | T100 PCC↑ | T100 RMSE↓ | T50 PCC↑ | T50 RMSE↓ |
|---|---|---|---|---|---|---|---|---|---|---|
| **TCGA-LUAD** | HE2RNA (Schmauch et al., 2020) | ResNet-50 | 0.104 | 0.513 | 0.117 | 0.508 | 0.142 | 0.500 | 0.152 | 0.498 |
| | tRNAsformer (Alsaafin et al., 2023) | UNI | 0.688 | 0.101 | 0.711 | 0.095 | 0.777 | 0.085 | 0.810 | 0.082 |
| | SEQUOIA (Pizurica et al., 2024) | ResNet-50 | 0.647 | 0.079 | 0.677 | 0.071 | 0.767 | 0.058 | 0.807 | 0.054 |
| | | UNI | 0.687 | 0.101 | 0.709 | 0.094 | 0.759 | 0.085 | 0.784 | 0.082 |
| | **RNA-FM (Ours)** | ResNet-50 | 0.653 | 0.076 | 0.678 | 0.069 | 0.763 | 0.060 | 0.808 | 0.054 |
| | | **UNI** | **0.729** | **0.074** | **0.756** | **0.066** | **0.834** | **0.054** | **0.865** | **0.050** |
| **TCGA-BRCA** | HE2RNA (Schmauch et al., 2020) | ResNet-50 | 0.067 | 0.411 | 0.076 | 0.406 | 0.095 | 0.397 | 0.104 | 0.395 |
| | tRNAsformer (Alsaafin et al., 2023) | UNI | 0.715 | 0.081 | 0.735 | 0.074 | 0.781 | 0.064 | 0.801 | 0.060 |
| | SEQUOIA (Pizurica et al., 2024) | ResNet-50 | 0.659 | 0.077 | 0.680 | 0.069 | 0.743 | 0.056 | 0.781 | 0.052 |
| | | UNI | 0.712 | 0.079 | 0.732 | 0.071 | 0.773 | 0.060 | 0.790 | 0.056 |
| | **RNA-FM (Ours)** | ResNet-50 | 0.674 | 0.082 | 0.697 | 0.072 | 0.760 | 0.061 | 0.788 | 0.055 |
| | | **UNI** | **0.744** | **0.072** | **0.766** | **0.064** | **0.824** | **0.052** | **0.856** | **0.048** |
| **TCGA-COAD** | HE2RNA (Schmauch et al., 2020) | ResNet-50 | 0.136 | 0.459 | 0.150 | 0.452 | 0.178 | 0.443 | 0.191 | 0.441 |
| | tRNAsformer (Alsaafin et al., 2023) | UNI | 0.716 | 0.069 | 0.757 | 0.062 | 0.836 | 0.052 | 0.860 | 0.049 |
| | SEQUOIA (Pizurica et al., 2024) | ResNet-50 | 0.708 | 0.069 | 0.764 | 0.061 | 0.864 | 0.050 | 0.894 | 0.047 |
| | | UNI | 0.698 | 0.073 | 0.733 | 0.066 | 0.813 | 0.056 | 0.843 | 0.053 |
| | **RNA-FM (Ours)** | ResNet-50 | 0.718 | 0.069 | 0.768 | 0.062 | 0.863 | 0.052 | 0.885 | 0.047 |
| | | **UNI** | **0.775** | **0.067** | **0.827** | **0.059** | **0.920** | **0.049** | **0.943** | **0.045** |

*Table 2.* Decision on the Gene Ontology (GO) aspect. GO-MF and GO-BP denote the molecular function and biological process aspects, respectively. We observe that GO-BP provides more informative prior knowledge for RNA-FM, yielding consistently better performance.

| GO Aspect | Spindle checkpoint signaling PCC↑ | RMSE↓ | Cell cycle DNA replication PCC↑ | RMSE↓ | Regulation of spindle checkpoint PCC↑ | RMSE↓ | Negative regulation of nuclear division PCC↑ | RMSE↓ |
|---|---|---|---|---|---|---|---|---|
| GO-MF | 0.623 | 0.296 | 0.638 | 0.236 | 0.603 | 0.293 | 0.606 | 0.368 |
| **GO-BP** | **0.682** | **0.254** | **0.668** | **0.217** | **0.667** | **0.243** | **0.664** | **0.231** |

these results demonstrate that RNA-FM provides a robust and accurate framework for genome-wide bulk RNA-seq prediction from WSIs across diverse tissue types.

### 4.3. Pathway-Level Analysis

Table 2 demonstrates the impact of different Gene Ontology (GO) aspects used as pathway priors within RNA-FM on pathway-level gene expression prediction. Across all examined functional categories, incorporating GO Biological Process (GO-BP) annotations consistently yields higher PCC and lower RMSE compared to using GO Molecular Function (GO-MF). This improvement is observed across diverse processes, and Table 2 lists the representative functional categories. These results indicate that GO-BP annotations provide a more organized prior for modeling bulk transcriptomic variation. Consequently, RNA-FM uses GO-BP as prior pathway knowledge, enabling more biologically meaningful genome-wide gene expression generation.

With the prior pathway knowledge GO-BP incorporated into RNA-FM, it consistently achieves higher pathway-level PCC than SEQUOIA, indicating the significance of

pathway-embedded structure in RNA-FM. Figure 3 demonstrates 4 representative pathway results comparing between RNA-FM and SEQUOIA methods. Specifically, RNA-FM increases PCC from 0.627 to 0.659 for antigen processing and presentation of exogenous peptide antigen, 0.632 to 0.673 for positive regulation of chromosome separation, and 0.626 to 0.682 for spindle checkpoint signaling. Notably, the largest improvement is observed for pancreatic A cell differentiation, where PCC improves substantially from 0.57 to 0.697. Overall, these results suggest RNA-FM provides more accurate pathway-level predictions across diverse biological processes.

### 4.4. Uncertainty Evaluation

To better characterize the uncertainty (variance) of generated samples by generative model RNA-FM, we generated 100 predictions per sample to assess calibration using empirical interval coverage and Gaussian NLL, along with the evaluation of the usefulness of uncertainty via the Spearman correlation between predictive variance and absolute error. The results are summarized in Table 11, which shows

*Table 3.* Comparison of generalization performance with baselines across datasets in CPTAC. The T1000, T500, T100, and T50 denote the top-1000, 500, 100, and 50 predictive genes, respectively. The **best results** are in bold; the second-best results are underlined.

| Dataset | Method | T1000 | | T500 | | T100 | | T50 | |
|---|---|---|---|---|---|---|---|---|---|
| | | PCC↑ | RMSE↓ | PCC↑ | RMSE↓ | PCC↑ | RMSE↓ | PCC↑ | RMSE↓ |
| CPTAC-LUAD | tRNAsformer (Alsaafin et al., 2023) | 0.311 | 0.115 | 0.349 | 0.103 | 0.424 | 0.085 | 0.447 | 0.079 |
| | SEQUOIA (Pizurica et al., 2024) | 0.334 | 0.112 | 0.369 | 0.100 | 0.433 | 0.081 | 0.456 | 0.075 |
| | **RNA-FM (Ours)** | **0.362** | **0.075** | **0.404** | **0.067** | **0.480** | **0.055** | **0.500** | **0.052** |
| CPTAC-BRCA | tRNAsformer (Alsaafin et al., 2023) | 0.380 | 0.111 | 0.417 | 0.099 | 0.481 | 0.079 | 0.504 | 0.073 |
| | SEQUOIA (Pizurica et al., 2024) | 0.375 | 0.110 | 0.411 | 0.098 | 0.472 | 0.080 | 0.491 | 0.074 |
| | **RNA-FM (Ours)** | **0.456** | **0.105** | **0.498** | **0.093** | **0.557** | **0.073** | **0.572** | **0.066** |
| CPTAC-COAD | tRNAsformer (Alsaafin et al., 2023) | 0.261 | 0.090 | 0.293 | 0.080 | 0.351 | 0.064 | 0.373 | 0.059 |
| | SEQUOIA (Pizurica et al., 2024) | 0.325 | 0.095 | 0.363 | 0.082 | 0.432 | 0.066 | 0.457 | 0.061 |
| | **RNA-FM (Ours)** | **0.396** | **0.075** | **0.439** | **0.067** | **0.515** | **0.054** | **0.543** | **0.050** |

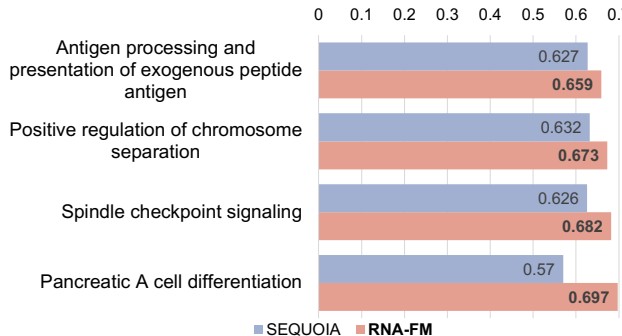

*Figure 3.* Comparison of Pathway-Level Results between RNA-FM and SEQUOIA. The values in the bars denote PCC.

nominal behavior at the 50%/80%/90% levels for different predictive gene sets, indicating RNA-FM's uncertainty estimates track the overall confidence level well. The consistent low Gaussian NLL further suggests that uncertainty estimation is sharp and reliable. Importantly, variance-error correlation remains consistently positive, indicating that RNA-FM assigns higher uncertainty to more difficult predictions. Overall, these results support that RNA-FM provides useful uncertainty estimates beyond point prediction.

### 4.5. Generalization Performance

Table 3 reports the external validation results on the CPTAC cohorts, evaluating the generalization ability of models trained on TCGA datasets across multiple cancer types. Across all three anatomical sites (CPTAC-LUAD, CPTAC-BRCA, and CPTAC-COAD), RNA-FM consistently achieves the highest PCC and lowest RMSE values for all predictive gene subsets, including the top-1000, 500, 100, and 50 predictive genes, and top-200 and 20 predictive genes demonstrated in Appendix Section E. Notably, the performance improvements are particularly significant for the most predictive gene subsets, e.g., the top-100 and top-50 gene subsets, indicating that RNA-FM preserves fine-grained transcriptomic signals under a significant domain shift. Compared with existing regression methods

such as tRNAsformer and SEQUOIA, RNA-FM demonstrates substantially improved robustness when transferring across cohorts with distinct experimental protocols. These results highlight the strong generalization capability of the proposed flow-matching generative framework and suggest that modeling the conditional gene expression distribution provides greater resilience to dataset shift.

### 4.6. Ablation Study

**Ablation on RNA-FM Architecture.** Table 4 presents an ablation study on the LUAD cohort to assess the contribution of key components in the RNA-FM framework, and the results of BRCA and COAD are shown in Appendix Section G. The full model consistently achieves the best performance across all gene subsets, from the top-1000 down to the top-20 predictive genes, demonstrating the effectiveness of the complete design. Without incorporating pathway-aware representations, we directly patch the 10 adjacent genes (in alphabetical order by gene name) for each token. Integrating the non-biological meaningful patchify rather than the pathway-aware patchify results in substantial degradation in both PCC and RMSE, particularly for smaller gene subsets (from top-100 to top-20 predictive genes), indicating that organizing genes into pathway-level patches is crucial for capturing biologically meaningful structure and improving predictive accuracy. Furthermore, eliminating the inter-pathway graph attention mechanism results in a consistent decrease in performance across all settings, highlighting its role in attending to functional dependencies between pathways. Overall, these results confirm that both pathway-aware representation incorporation and inter-pathway interaction modules contribute independently and synergistically to RNA-FM performance, validating the necessity of each component for accurate and robust genome-wide bulk RNA-seq prediction.

**Ablation on Learning Rate.** Table 5 demonstrates an ablation study on the learning rate used to train RNA-FM. Across all gene subsets, a learning rate of $1e^{-4}$ consistently

*Table 4.* Ablation study on RNA-FM architecture on TCGA-LUAD. The **best results** are in bold.

| Method | T1000 | | T500 | | T200 | | T100 | | T50 | | T20 | |
|---|---|---|---|---|---|---|---|---|---|---|---|---|
| | PCC↑ | RMSE↓ | PCC↑ | RMSE↓ | PCC↑ | RMSE↓ | PCC↑ | RMSE↓ | PCC↑ | RMSE↓ | PCC↑ | RMSE↓ |
| **Full Framework** | **0.729** | **0.074** | **0.756** | **0.066** | **0.798** | **0.059** | **0.834** | **0.054** | **0.865** | **0.050** | **0.897** | **0.046** |
| *w/o* pathway-aware patchify | 0.679 | 0.097 | 0.701 | 0.082 | 0.745 | 0.076 | 0.796 | 0.070 | 0.831 | 0.061 | 0.866 | 0.054 |
| *w/o* inter-pathway graph attention | 0.716 | 0.076 | 0.743 | 0.068 | 0.784 | 0.060 | 0.820 | 0.055 | 0.852 | 0.051 | 0.886 | 0.047 |

*Table 5.* Ablation study on the learning rate for training. The **best results** are in bold.

| Learning rate | T1000 | | T500 | | T100 | | T50 | |
|---|---|---|---|---|---|---|---|---|
| | PCC↑ | RMSE↓ | PCC↑ | RMSE↓ | PCC↑ | RMSE↓ | PCC↑ | RMSE↓ |
| $2e^{-4}$ | 0.701 | 0.083 | 0.731 | 0.080 | 0.801 | 0.068 | 0.855 | 0.060 |
| $1e^{-4}$ | **0.729** | **0.074** | **0.756** | **0.066** | **0.834** | **0.054** | **0.865** | **0.050** |

*Table 6.* Ablation study on the interpolant for sampling. The **best results** are in bold.

| Interpolant | T1000 | | T500 | | T100 | | T50 | |
|---|---|---|---|---|---|---|---|---|
| | PCC↑ | RMSE↓ | PCC↑ | RMSE↓ | PCC↑ | RMSE↓ | PCC↑ | RMSE↓ |
| Logistic | 0.709 | 0.081 | 0.738 | 0.079 | 0.823 | 0.059 | 0.859 | 0.058 |
| Linear | **0.729** | **0.074** | **0.756** | **0.066** | **0.834** | **0.054** | **0.865** | **0.050** |

*Table 7.* Ablation study on the CFG scale $s$. $s = 1$ denotes that a classifier-free guidance mechanism is NOT employed in the inference (sampling) process. The **best results** are in bold.

| CFG scale | T1000 | | T500 | | T100 | | T50 | |
|---|---|---|---|---|---|---|---|---|
| | PCC↑ | RMSE↓ | PCC↑ | RMSE↓ | PCC↑ | RMSE↓ | PCC↑ | RMSE↓ |
| $s = 1$ | 0.697 | 0.098 | 0.0719 | 0.086 | 0.802 | 0.071 | 0.840 | 0.067 |
| $s = 2$ | **0.729** | **0.074** | **0.756** | **0.066** | **0.834** | **0.054** | **0.865** | **0.050** |
| $s = 3$ | 0.718 | 0.079 | 0.742 | 0.074 | 0.827 | 0.056 | 0.851 | 0.055 |

error, suggesting a favorable balance between conditional fidelity and generative diversity. Instead, increasing the guidance strength further to $s = 3$ leads to a slight performance decrease, particularly for larger gene subsets. Overall, these results demonstrate that classifier-free guidance is an effective mechanism for strengthening histopathology-conditioned transcriptomic generation in RNA-FM.

## 5. Conclusion

In this work, we introduced RNA-FM, a pathway-aware flow-matching generative framework for genome-wide bulk RNA-seq prediction from WSIs. By formulating transcriptomic imputation as a continuous-time conditional transport problem, RNA-FM moves beyond deterministic regression and models the conditional distribution of gene expression given tissue morphology, thereby capturing biologically meaningful variability. To make genome-scale generation both structured and tractable, RNA-FM tokenizes genes into pathway-level units and learns dependencies across pathways, enabling biologically interpretable representation. Extensive experiments demonstrate that RNA-FM consistently outperforms existing methods. Across settings, RNA-FM better captures both inter-patient and intra-tumoral heterogeneity while maintaining biological meaningfulness, supporting its potential as a practical foundation for morphology-conditioned transcriptomic generation.

**Limitations and Future Work.** While RNA-FM generalizes across the anatomical sites studied, an important next step is to incorporate additional datasets spanning more anatomical sites and disease contexts to move toward a truly tissue-agnostic, genome-wide RNA-seq imputation generative model. Beyond expanding site diversity, future work could extend the framework to other transcriptomic modalities, e.g., single-cell or spatially resolved measurements, to further enhance controllability and uncertainty quantification for downstream clinical and biological applications.

outperforms a higher rate of $2e^{-4}$, achieving higher PCC and lower RMSE. The performance gain is especially significant for highly predictive gene subsets, i.e., from top-100 to top-50, suggesting that a modestly lower learning rate yields more stable optimization and better convergence for the flow-matching generative objective. Thus, we set the learning rate to $1e^{-4}$ as the default for all experiments.

**Ablation on Interpolant.** To evaluate the choice of interpolant in the RNA-FM flow-matching generative model, we conduct experiments with a Logistic and a Linear interpolant. Table 6 compares different interpolants used in RNA-FM for defining the probability transport path. The Linear interpolant consistently outperforms the Logistic alternative across all datasets we used, yielding higher PCC and lower RMSE. Thus, linear interpolation provides a more stable and effective supervision signal for learning the velocity field in high-dimensional gene expression space, supporting the use of a Linear interpolant in RNA-FM and aligning with observations that uniform transport paths facilitate reliable optimization in flow-matching generative models.

**Ablation on CFG.** Table 7 evaluates the effect of classifier-free guidance (CFG) during sampling by varying the guidance scale $s$. When CFG is disabled ($s = 1$), the model exhibits consistently lower PCC and higher RMSE across all gene subsets, indicating that unconditional sampling underutilizes the morphology conditioning signal. Introducing CFG substantially improves performance, with a moderate guidance scale of $s = 2$ yielding the best overall results across the top 1000 to the top 50 predictive genes. This setting enhances correlation while reducing prediction

## Impact Statement

This research study was conducted retrospectively using human subject data made available in open access from the Genomic Data Commons (GDC) portal (https://portal.gdc.cancer.gov) and The Cancer Imaging Archive (TCIA) (https://www.cancerimagingarchive.net). Ethical approval was not required, as confirmed by the license attached to the open-access data.

This paper presents work whose goal is to advance the field of Machine Learning for Biomedical Transcriptomics Research, and the source code is released at https://github.com/YXSong000/RNA-FM to support further research by the community. There are many potential societal consequences of our work, none of which we feel must be specifically highlighted here.

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

## A. Proof for Probability Flow ODE for Conditional Gene Expression Transport

Following the studies (Lipman et al., 2023; Ma et al., 2024), we consider a continuous-time interpolation between a target gene expression sample and Gaussian noise. Given:

$$x_1 \sim p_{\text{gene}}(\cdot \mid y), \qquad x_0 \sim p_0 = \mathcal{N}(0, I_G), \tag{15}$$

where $x_1 \in \mathbb{R}^G$ denotes a genome-wide bulk RNA-seq expression vector conditioned on histopathology features $y$, and $G$ is the number of genes.

We define a time-dependent random variable $\{x_t\}_{t \in [0,1]}$ as:

$$x_t = \alpha(t) \, x_1 + \sigma(t) \, x_0, \qquad t \in [0, 1], \tag{16}$$

where $\alpha(t)$ and $\sigma(t)$ are smooth scalar functions satisfying the boundary conditions:

$$\alpha(0) = 0, \ \sigma(0) = 1, \qquad \alpha(1) = 1, \ \sigma(1) = 0.$$

Let $p_t(x \mid y)$ denote the conditional probability density function (PDF) of $x_t$ given $y$. Its characteristic function is defined as:

$$\hat{p}_t(k \mid y) = \int_{\mathbb{R}^G} e^{ik^\top x} \, p_t(x \mid y) \, \mathrm{d}x = \mathbb{E}\left[ e^{ik^\top x_t} \mid y \right], \tag{17}$$

where the expectation is taken over the joint distribution of $x_0$ and $x_1$.

Specifically, the default setting in RNA-FM utilizes a Linear interpolant, sampling the diffusion time steps uniformly from $[0, 1]$ with the choice of $\alpha_t = 1 - t$, $\sigma_t = t$.

### A.1. Time Evolution of the Characteristic Function

Taking the time derivative of Equation (17) and applying the tower property of conditional expectation yields:

$$\begin{aligned}
\partial_t \hat{p}_t(k \mid y) &= ik^\top \mathbb{E}\left[ \dot{x}_t \, e^{ik^\top x_t} \mid y \right] \\
&= ik^\top \mathbb{E}_{x \sim p_t(\cdot \mid y)}\left[ \mathbb{E}[\dot{x}_t \mid x_t = x, \, y] \, e^{ik^\top x} \right].
\end{aligned} \tag{18}$$

From Equation (16), we have:

$$\dot{x}_t = \dot{\alpha}(t) \, x_1 + \dot{\sigma}(t) \, x_0.$$

We define the conditional velocity field:

$$v(x, t \mid y) \triangleq \mathbb{E}[\dot{x}_t \mid x_t = x, \, y] = \dot{\alpha}(t) \, \mathbb{E}[x_1 \mid x_t = x, \, y] + \dot{\sigma}(t) \, \mathbb{E}[x_0 \mid x_t = x, \, y]. \tag{19}$$

Substituting Equation (19) into Equation (18) yields:

$$\partial_t \hat{p}_t(k \mid y) = ik^\top \int_{\mathbb{R}^G} v(x, t \mid y) \, e^{ik^\top x} \, p_t(x \mid y) \, \mathrm{d}x. \tag{20}$$

### A.2. Derivation of the Transport Equation

Equation (20) can be rewritten as:

$$\begin{aligned}
\int_{\mathbb{R}^G} e^{ik^\top x} \, \partial_t p_t(x \mid y) \, \mathrm{d}x &= \int_{\mathbb{R}^G} v(x, t \mid y)^\top \nabla_x\left( e^{ik^\top x} \right) p_t(x \mid y) \, \mathrm{d}x \\
&= -\int_{\mathbb{R}^G} \nabla_x \cdot (v(x, t \mid y) \, p_t(x \mid y)) \, e^{ik^\top x} \, \mathrm{d}x,
\end{aligned} \tag{21}$$

where the second equality follows from integration by parts, assuming sufficient decay of $p_t(x \mid y)$ at infinity. Here, $\nabla_x \cdot (\cdot)$ denotes the divergence operator in $\mathbb{R}^G$.

By uniqueness of the Fourier transform, Equation (21) implies that $p_t(x \mid y)$ satisfies the continuity (transport) equation:

$$\partial_t p_t(x \mid y) + \nabla_x \cdot (v(x, t \mid y) \, p_t(x \mid y)) = 0. \tag{22}$$

The transport Equation (22) admits a characteristic solution given by the conditional probability flow ordinary differential equation:

$$\frac{\mathrm{d}X_t}{\mathrm{d}t} = v(X_t, t \mid y), \tag{23}$$

whose solution preserves the marginal distribution $X_t \sim p_t(\cdot \mid y)$. Integrating Equation (23) backward in time from $X_0 \sim \mathcal{N}(0, I_G)$ yields samples from the conditional gene expression distribution $p_{\text{gene}}(x \mid y)$.

## B. Additional Implementation Details on RNA-FM Architecture

*Table 8.* Dimension summary for each model component. $B$ is batch size, $G$ is the number of genes, $P$ is the number of pathways, $|P_i|$ is the size of pathway $i$, $|B|$ is the background gene-set size, $K$ is the number of condition tokens (clusters), $d$ is the condition feature dimension, and $h$ is the latent hidden dimension.

| Component | Role | Input $\rightarrow$ Output dimensions |
|---|---|---|
| Time embedding | Embed continuous flow time $t$ | $t \in \mathbb{R}^B \;\rightarrow\; \mathbf{t} \in \mathbb{R}^{B \times h}$ |
| Gene-to-pathway tokens | Aggregate gene expression into pathway-level tokens with a background gene token | $\mathbf{x} \in \mathbb{R}^{B \times G} \;\rightarrow\; \mathbf{z} \in \mathbb{R}^{B \times P \times h}, \; \mathbf{z}_{\text{bg}} \in \mathbb{R}^{B \times 1 \times h}$ |
| Pathway graph block | Graph-constrained inter-pathway dependency attention | $\mathbf{z} \in \mathbb{R}^{B \times P \times h} \;\rightarrow\; \mathbf{z} \in \mathbb{R}^{B \times P \times h}$ |
| WSI Cluster-token | Model the WSI cluster features and pooling | $\mathbf{y} \in \mathbb{R}^{B \times K \times d} \;\rightarrow\; \mathbf{y} \in \mathbb{R}^{B \times K \times d} \;\rightarrow\; \bar{\mathbf{y}} \in \mathbb{R}^{B \times d}$ |
| Condition projection | Project condition into the latent space | $\bar{\mathbf{y}} \in \mathbb{R}^{B \times d} \;\rightarrow\; \mathbf{y}_{\text{emb}} \in \mathbb{R}^{B \times h}$ |
| Joint conditioning | Combine time and condition embeddings | $\mathbf{t} \in \mathbb{R}^{B \times h}, \; \mathbf{y}_{\text{emb}} \in \mathbb{R}^{B \times h} \;\rightarrow\; \mathbf{c} \in \mathbb{R}^{B \times h}$ |
| Conditional DiT backbone | Iterative conditional transformation of pathway tokens | $\mathbf{z} \in \mathbb{R}^{B \times P \times H}, \; \mathbf{c} \in \mathbb{R}^{B \times h} \;\rightarrow\; \mathbf{z} \in \mathbb{R}^{B \times P \times h}$ |
| Pathway-specific heads | Predict gene values within each pathway | $\mathbf{z} \in \mathbb{R}^{B \times P \times h}, \; \mathbf{c} \in \mathbb{R}^{B \times h} \;\rightarrow\; \{\hat{\mathbf{x}}_i \in \mathbb{R}^{B \times |P_i|}\}_{i=1}^{P}$ |
| Background gene head | Predict values for background-gene set | $\mathbf{z}_{\text{bg}} \in \mathbb{R}^{B \times 1 \times h}, \; \mathbf{c} \in \mathbb{R}^{B \times h} \;\rightarrow\; \hat{\mathbf{x}}_{\text{bg}} \in \mathbb{R}^{B \times |B|}$ |
| Pathway-to-gene reconstruction | Combine pathway and background predictions into full gene-space output | $\{\hat{\mathbf{x}}_i \in \mathbb{R}^{B \times |P_i|}\}_{i=1}^{P}, \; \hat{\mathbf{x}}_{\text{bg}} \in \mathbb{R}^{B \times |B|} \;\rightarrow\; \hat{\mathbf{x}} \in \mathbb{R}^{B \times G}$ |
| Classifier-free guidance | Guided sampling using conditional and unconditional predictions | $\hat{\mathbf{x}}_{\text{cond}} \in \mathbb{R}^{B \times G}, \; \hat{\mathbf{x}}_{\text{uncond}} \in \mathbb{R}^{B \times G} \;\rightarrow\; \hat{\mathbf{x}}_{\text{guided}} \in \mathbb{R}^{B \times G}$ |

Table 8 summarizes the implementation of RNA-FM by mapping each major model component to its functional role in the framework, together with the corresponding tensor dimensions and the main operations performed. The table follows the forward pass from gene-level inputs to pathway tokens, incorporates pathway graph–constrained interactions and WSI-derived conditioning, and then decodes pathway-level predictions back into the full gene space. We also include the classifier-free guidance mechanism used at sampling.

## C. Additional Implementation Details on Evaluation Metrics

Throughout the experiments, we utilize the Pearson Correlation Coefficient (PCC) and the Root Mean Squared Error (RMSE) as evaluation metrics. Specifically, they are calculated by:

$$\text{PCC} = \frac{\sum_{i=1}^{N}(x_i - \bar{x})(\hat{x}_i - \bar{\hat{x}})}{\sqrt{\sum_{i=1}^{N}(x_i - \bar{x})^2}\sqrt{\sum_{i=1}^{N}(\hat{x}_i - \bar{\hat{x}})^2}} \text{ and} \tag{24}$$

$$\text{RMSE} = \sqrt{\frac{1}{N}\sum_{i=1}^{N}(x_i - \hat{x}_i)^2}. \tag{25}$$

In this study, we report both the mean PCC and the mean RMSE for top-1000, 500, 200, 100, 50, and 20 highly predictive genes. Specifically, PCC and RMSE are computed for each gene across all samples in each dataset, and various highly predictive gene sets are identified by cross-validating their averaged out-of-fold test PCC performance across all folds in transcriptomic prediction.

## D. Additional Cross-validation Experiments Results

*Table 9.* Additional comparison of cross-validation experiments with baselines across datasets in TCGA. The T200 and T20 denote the top-200 and top-20 predictive genes, respectively. The **best results** are in bold; the second-best results are underlined.

| Method | Feature Extractor | TCGA-LUAD | | | | TCGA-BRCA | | | | TCGA-COAD | | | |
|---|---|---|---|---|---|---|---|---|---|---|---|---|---|
| | | T200 | | T20 | | T200 | | T20 | | T200 | | T20 | |
| | | PCC↑ | RMSE↓ | PCC↑ | RMSE↓ | PCC↑ | RMSE↓ | PCC↑ | RMSE↓ | PCC↑ | RMSE↓ | PCC↑ | RMSE↓ |
| HE2RNA | ResNet-50 | 0.132 | 0.503 | 0.163 | 0.494 | 0.086 | 0.400 | 0.116 | 0.391 | 0.167 | 0.447 | 0.214 | 0.437 |
| tRNAsformer | UNI | 0.746 | 0.088 | 0.850 | 0.078 | 0.761 | 0.068 | 0.831 | 0.057 | 0.807 | 0.055 | 0.885 | 0.046 |
| SEQUOIA | ResNet-50 | 0.726 | 0.063 | 0.850 | 0.049 | 0.712 | 0.061 | 0.837 | 0.047 | 0.828 | 0.054 | 0.924 | 0.044 |
| | UNI | 0.737 | 0.088 | 0.817 | 0.078 | 0.756 | 0.064 | 0.814 | 0.052 | 0.781 | 0.060 | 0.874 | 0.050 |
| RNA-FM | ResNet-50 | 0.729 | 0.064 | 0.827 | 0.054 | 0.727 | 0.061 | 0.840 | 0.046 | 0.830 | 0.054 | 0.910 | 0.047 |
| | UNI | **0.798** | **0.059** | **0.897** | **0.046** | **0.798** | **0.057** | **0.908** | **0.043** | **0.888** | **0.052** | **0.964** | **0.042** |

As complement to Table 1, Table 9 reports additional cross-validation results on TCGA datasets for the top-200 and top-20 predictive gene subsets across LUAD, BRCA, and COAD. Overall, RNA-FM consistently achieves the strongest performance, particularly when combined with the UNI feature extractor, yielding the highest PCC and lowest RMSE across all cancer types and gene subsets. The gains are especially significant for smaller gene subsets, such as the top-20 predictive genes, where accurate modeling of high-variance, biologically informative genes is most challenging. Compared to deterministic regression methods such as HE2RNA and tRNAsformer, RNA-FM achieves markedly higher correlation and lower error, reflecting the benefits of probabilistic modeling. Relative to SEQUOIA, RNA-FM further improves both PCC and RMSE in nearly all settings, highlighting the complementary advantages of flow-matching generative modeling and pathway-aware representations. These results reinforce the robustness of RNA-FM across different anatomical sites.

## E. Additional Generalization Performance

*Table 10.* Comparison of generalization performance with baselines across datasets in CPTAC. The T200 and T20 denote the top-200 and top-20 predictive genes, respectively. The **best results** are in bold.

| Method | CPTAC-LUAD | | | | CPTAC-BRCA | | | | CPTAC-COAD | | | |
|---|---|---|---|---|---|---|---|---|---|---|---|---|
| | T200 | | T20 | | T200 | | T20 | | T200 | | T20 | |
| | PCC↑ | RMSE↓ | PCC↑ | RMSE↓ | PCC↑ | RMSE↓ | PCC↑ | RMSE↓ | PCC↑ | RMSE↓ | PCC↑ | RMSE↓ |
| tRNAsformer (Alsaafin et al., 2023) | 0.396 | 0.091 | 0.467 | 0.073 | 0.456 | 0.086 | 0.533 | 0.066 | 0.327 | 0.070 | 0.402 | 0.054 |
| SEQUOIA (Pizurica et al., 2024) | 0.409 | 0.088 | 0.480 | 0.071 | 0.449 | 0.086 | 0.516 | 0.066 | 0.404 | 0.072 | 0.485 | 0.056 |
| **RNA-FM (Ours)** | **0.454** | **0.060** | **0.520** | **0.048** | **0.537** | **0.081** | **0.586** | **0.059** | **0.485** | **0.059** | **0.571** | **0.045** |

As a complement to Table 3, Table 10 evaluates the generalization ability of models pretrained on TCGA by inference on independent CPTAC cohorts across CPTAC-LUAD, CPTAC-BRCA, and CPTAC-COAD. Across all cancer types and gene subsets, RNA-FM consistently achieves the highest PCC and lowest RMSE, outperforming both tRNAsformer and SEQUOIA. Notably, the performance improvements are particularly pronounced for the top-20 predictive genes, indicating that RNA-FM better preserves predictive accuracy for highly informative genes under domain shift. These results demonstrate that the proposed flow-matching framework learns transferable, morphology-conditioned transcriptomic representations that generalize beyond the training distribution. Overall, the strong performance on CPTAC validates the robustness and cross-cohort generalization capability of RNA-FM for genome-wide RNA-seq prediction.

## F. Additional Uncertainty Evaluation

*Table 11.* Quantitative uncertainty evaluation results. The T1000, T500 and T200 denote the top-1000, top-500 and top-200 predictive genes, respectively.

| Gene set | 50% interval coverage | 80% interval coverage | 90% interval coverage | Gaussian NLL ↓ | Spearman corr(var, abs. error) ↑ |
|---|---|---|---|---|---|
| T1000 | 0.521 | 0.766 | 0.834 | 1.433 | 0.646 |
| T500 | 0.550 | 0.791 | 0.850 | 0.652 | 0.597 |
| T200 | 0.586 | 0.819 | 0.869 | -0.521 | 0.521 |

As complement to Section 4.4, Table 11 provides a quantitative uncertainty evaluation result on TCGA-BRCA and suggests that the generated variance is meaningful rather than arbitrary. The empirical coverage is reasonably aligned with nominal levels at the 50%/80%/90% levels for different predictive gene sets, indicating RNA-FM's uncertainty estimates track the overall confidence level well. The consistent low Gaussian NLL further suggests that uncertainty estimation is sharp and reliable. Importantly, variance-error correlation remains consistently positive across all subsets, indicating that samples with larger variance tend to correspond to harder or more ambiguous predictions, which is consistent with our interpretation of RNA-FM as modeling a conditional distribution $p(x \mid y)$ rather than producing a deterministic point estimate. Overall, these results support that RNA-FM provides useful uncertainty estimates beyond point prediction.

## G. Additional Ablation Study on RNA-FM Architecture

*Table 12.* Ablation study on RNA-FM architecture on TCGA-BRCA. The **best results** are in bold.

| Method | T1000 | | T500 | | T200 | | T100 | | T50 | | T20 | |
|---|---|---|---|---|---|---|---|---|---|---|---|---|
| | PCC↑ | RMSE↓ | PCC↑ | RMSE↓ | PCC↑ | RMSE↓ | PCC↑ | RMSE↓ | PCC↑ | RMSE↓ | PCC↑ | RMSE↓ |
| **Full Framework** | **0.744** | **0.072** | **0.766** | **0.064** | **0.798** | **0.057** | **0.824** | **0.052** | **0.856** | **0.048** | **0.908** | **0.043** |
| *w/o* pathway-aware patchify | 0.708 | 0.090 | 0.724 | 0.084 | 0.759 | 0.073 | 0.789 | 0.063 | 0.818 | 0.059 | 0.846 | 0.048 |
| *w/o* inter-pathway graph attention | 0.736 | 0.073 | 0.757 | 0.065 | 0.786 | 0.057 | 0.814 | 0.052 | 0.849 | 0.048 | 0.904 | 0.044 |

*Table 13.* Ablation study on RNA-FM architecture on TCGA-COAD. The **best results** are in bold.

| Method | T1000 | | T500 | | T200 | | T100 | | T50 | | T20 | |
|---|---|---|---|---|---|---|---|---|---|---|---|---|
| | PCC↑ | RMSE↓ | PCC↑ | RMSE↓ | PCC↑ | RMSE↓ | PCC↑ | RMSE↓ | PCC↑ | RMSE↓ | PCC↑ | RMSE↓ |
| **Full Framework** | **0.775** | **0.067** | **0.827** | **0.059** | **0.888** | **0.052** | **0.920** | 0.049 | **0.943** | **0.045** | **0.964** | **0.042** |
| *w/o* pathway-aware patchify | 0.753 | 0.076 | 0.801 | 0.069 | 0.857 | 0.066 | 0.887 | 0.058 | 0.918 | 0.053 | 0.946 | 0.049 |
| *w/o* inter-pathway graph attention | 0.770 | 0.067 | 0.822 | 0.060 | 0.884 | 0.053 | 0.917 | **0.049** | 0.941 | 0.046 | 0.964 | 0.043 |

As a complement to Table 4, Table 12 demonstrates that the full RNA-FM framework achieves the best performance on BRCA across all token settings from top-1000 to top-20 predictive genes, consistently yielding the highest PCC and lowest RMSE. Removing the pathway-aware patchify results in the largest degradation, with notably lower correlations and higher errors across, e.g., at T1000, PCC drops from 0.744 to 0.708, and RMSE increases from 0.072 to 0.090, indicating that pathway-based gene tokenization is a key contributor. In contrast, removing inter-pathway graph attention results in a smaller but still consistent decline, e.g., at T1000, PCC 0.736 vs. 0.744; at T20, PCC 0.904 vs. 0.908, suggesting that modeling inter-pathway dependencies provides additional gains beyond pathway-aware patchify.

As a complement to Table 4, Table 13 presents the architecture ablation on COAD and shows that the full RNA-FM framework delivers the strongest overall performance across all token settings, from top-1000 to top-20 predictive genes, achieving the highest PCC and lowest RMSE in nearly every case. Removing the pathway-aware patchify consistently yields the largest decrease in correlation and increases error, e.g., at T500, PCC decreases from 0.827 to 0.801, and RMSE rises from 0.059 to 0.069, highlighting the importance of pathway-based gene tokenization. In contrast, removing inter-pathway graph attention results in a slight degradation relative to the full model, and it ties the best RMSE at T100 (0.049) while remaining slightly worse elsewhere, e.g., at T20, RMSE 0.043 vs. 0.042.

