# OpenReview forum: "RNA-FM: Flow-Matching Generative Model for Genome-wide RNA-Seq Prediction"
_ICML.cc/2026/Conference — ICML 2026 regular_

### Official Review · Reviewer_SZFU · 2026-02-14

**Soundness:** 4
**Presentation:** 3
**Significance:** 3
**Originality:** 4
**Overall Recommendation:** 4
**Confidence:** 5

**Summary:**

The paper presents RNA-FM. The authors design it as a generative model to predict bulk RNA-seq profiles from WSIimages. The method treats transcriptome prediction as a continuous transport problem. The model learns a velocity field that maps samples from a simple prior to the real gene expression distribution. The mapping is conditioned on image features extracted from pathology slides. The authors also add pathway structure into the generative process. They claim this design helps with biological meaning and scaling. The experiments use multiple tissue types and external cohorts. The results show that RNA-FM improves over previous image-to-expression models. The model also aims to capture both patient-level and spatial heterogeneity inside tumors.

**Compliance With Llm Reviewing Policy:**

Affirmed.

**Key Questions For Authors:**

1. The authors should clarify how they encode pathway structure clearly.
2. The authors should show do several ablation studies if they remove this component.
3. I am curious how RNA-FM performs compared with integration of prediction of single-cell-level or single-cell-level spatial transcriptomics. The authors could do several experiments about this.

**Limitations:**

The paper does not explicitly discuss limitations. We suggest the authors address potential issues such as dependence on pathway annotations, scalability to very large gene sets, and the assumption that image patches fully determine bulk expression. A clear discussion of cases where image features do not predict expression well would strengthen the manuscript.

**Strengths And Weaknesses:**

The method introduces a new direction by using flow-matching for transcriptome prediction. The design moves beyond single point estimates and instead models a full distribution. The authors include biological pathway information, and this addition may improve biological consistency. The experiments include external validation, which increases credibility. The results show performance gains over the compared baselines. The model also explicitly addresses tumor heterogeneity, which is important in clinical data.

The model predicts bulk RNA-seq only and does not resolve single-cell expression or spatial transcriptomics. The pathway constraint may bias the model toward known genes and pathways. The comparison does not clearly include other recent generative methods.

---

> ### Author Rebuttal · Authors · 2026-03-31
>
> We thank you for recognizing the novelty of applying flow matching to transcriptome prediction, the biological pathway integration, and the effectiveness of external validation, and for thoughtful suggestions. Please find our detailed responses below.
>
> **1. Clarify on pathway structure**
>
> We appreciate you pointing this out. As shown in *Section 3.2* of our paper, the pathway-aware representation is described in detail: genes are grouped into Gene Ontology(GO)-based pathway sets, uncovered genes are handled through an additional background gene set, pathway tokens are learned with pathway-specific embeddings, inter-pathway dependencies are modeled using a pathway graph derived from overlapping genes across pathways, and gene expression reconstruction is performed by overlap-aware averaging. We would like to explain it more clearly and directly in the revision.
>
> **2. Clarify on the pathway component**
>
> According to *Section 4.3*, from the perspective of the **choice/selection of pathway component**, even though RNA-FM is able to handle various Gene Ontology (GO) aspects with different pathway granularities/overlapping, *Table 2* in our paper showed that GO-BP (contains 1931 pathways and 4314 background genes) gives better pathway-level performance than GO-MF (contains 382 pathways and 3645 background genes). Therefore, we choose GO-BP as the pathway prior to RNA-FM.
>
> According to *Section 4.5*, from the perspective of **architecture effectiveness of the pathway component**, *Table 4* in our paper illustrates the architecture ablation on the pathway component, showing that removing our pathway-aware patchification or inter-pathway graph attention degrades predictive performance.
>
> Thanks to you for pointing this out. We will point to these ablations more explicitly in the revision to make the contribution of pathway component easier to follow.
>
> **3. Additional spatial transcriptomics prediction results**
>
> We fully agree that this is an interesting and valuable future direction, as we discussed in *Limitations and Future Work* of *Section 5 Conclusion* in our paper. While this is beyond the main scope of the present work, which focuses on genome-wide **bulk** RNA-seq prediction from WSIs, we would nevertheless like to share preliminary evidence that RNA-FM can transfer to tile-level spatial transcriptomics (ST) inference:
>
> |Model|GNAS (PCC↑)|FASN (PCC↑)|
> |:--- |:---:|:---:|
> |SEQUOIA|0.398|0.480|
> |**RNA-FM**|**0.431**|**0.562**|
>
> We applied RNA-FM to tile-level spatial gene expression inference on the ST dataset HER2ST`[1]`, which contains 36 slides. The results show the average PCC across slides for two well-established cancer-relevant biomarkers, GNAS`[2]` and FASN`[3]`, which are among the highly predictive genes across datasets. These results suggest that RNA-FM has promising potential to generalize beyond bulk-level prediction and adapt to ST settings as well.
>
> [1] Andersson, A., et al. (2021). Spatial deconvolution of HER2-positive breast cancer delineates tumor-associated cell type interactions. *Nature communications*.
>
> [2] Afolabi, H. A., et al. (2022). A GNAS gene mutation's independent expression in the growth of colorectal cancer: a systematic review and meta-analysis. *Cancers*.
>
> [3] Xiao, Y., et al. (2024). The implications of FASN in immune cell biology and related diseases. *Cell death & disease*.
>
> **4. Clarify on limitations**
>
> For the **dependence on pathway annotations** and **scalability to large gene set**, please refer to *2. Clarify on the pathway component*, *Table 2* in the paper demonstrates that RNA-FM is able to handle various Gene Ontology (GO) aspects (GO-BP or GO-MF) across different pathway granularities/overlapping with various number of genes.
>
> Besides, we appreciate you raising the **residual ambiguity in histology-to-bulk expression mapping**. We will include the metrics, such as NLL, interval coverage (IC), and variance-abs. error Spearman correlation, to better reflect the meaningfulness of uncertianty.
>
> |Gene|50% IC|80% IC|90% IC|NLL↓|Corr↑|
> |:--- |:---:|:---:|:---:|:---:|:---:|
> |T1000|0.521|0.766|0.834|1.433|0.646|
> |T500|0.550|0.791|0.850|0.652|0.597|
> |T200|0.586|0.819|0.869|-0.521|0.521|
>
> The results show nominal behavior at the 50%/80%/90% levels for different predictive gene sets, indicating RNA-FM‘s uncertainty estimates track the overall confidence level well. The consistent low Gaussian NLL further suggests that uncertainty estimation is sharp and reliable. Importantly, variance-error correlation remains consistently positive, indicating that RNA-FM assigns higher uncertainty to more difficult predictions. Overall, these results support that RNA-FM provides useful uncertainty estimates beyond point prediction, while residual ambiguity in histology-to-bulk expression mapping is partially explained. We will mention and clarify the residual ambiguity in histology-to-bulk expression in the section of *Limitations* in the revision.

---

> > ### Author Rebuttal · Reviewer_SZFU · 2026-04-05
> >
> > The authors have resolved all of my concerns.

---

> > > ### Author Response · Authors · 2026-04-06
> > >
> > > We are very glad to hear that your concerns have been fully resolved. We appreciate your recognition of our rebuttal and all of your constructive suggestions, which are very helpful for further improving the paper. We will incorporate them into the revised version. Thank you again for your time and support.

---

### Official Review · Reviewer_teYn · 2026-02-27

**Soundness:** 3
**Presentation:** 3
**Significance:** 2
**Originality:** 3
**Overall Recommendation:** 4
**Confidence:** 3

**Summary:**

This submission introduces RNA-FM, a flow matching method to generate bulk RNA-seq for a given pathology patch. A key design choice is a gene-ontology pathway token to include an inductive bias for prior biological knowledge. Across various cancer datasets, the authors show that RNA-FM outperforms baseline methods and also design a careful ablation on chosen parameters.

**Compliance With Llm Reviewing Policy:**

Affirmed.

**Key Questions For Authors:**

See weaknesses.

**Limitations:**

Yes

**Strengths And Weaknesses:**

Strengths:
- The modelling choice makes sense and the addressed problem is important
- The paper is clearly written and easy to follow
- The pathway tokenization is an interesting, biologically grounded novelty

Weaknesses:
- The method is motivated as an uncertainty-aware method, but the evaluation is mostly focused on PCC/RMSE on point predictions or summary statistics. This strongly weakens the conclusion in my perspective. I'd recommend comparing how the plethora of generative models for single/bulk RNA seq evaluates the goodness of generated vectors. I think there may be easy additions that give a fuller picture of the performance.
- The submission as it is heavily claims heterogeneity and uncertainty as objectives, but doesn't really do much with it. A stronger evaluation of the uncertainty and heterogeneity could be helpful, what are the downstream use cases? Or, if not possible, I suggest toning this aspect down.

---

> ### Author Rebuttal · Authors · 2026-03-31
>
> We thank you for acknowledging and recognizing the importance of our problem, the clarity of the presentation, and the novelty of biologically grounded pathway tokenization, and for giving constructive suggestions. Please find our detailed responses below.
>
> **1. Additional Uncertainty evaluation**
>
> We appreciate this thoughtful suggestion, which can help improve our paper. To better characterize the uncertainty (variance) of generated samples, we generated 100 predictions per sample to assess calibration using empirical interval coverage and Gaussian NLL, along with the evaluation of the usefulness of uncertainty via the Spearman correlation between predictive variance and absolute error. Please kindly refer to the table below for the results.
>
> | Gene set | 50% interval coverage | 80% interval coverage | 90% interval coverage | Gaussian NLL ↓ | Spearman corr(var, abs. error) ↑ |
> |:-------- |:---------------------:|:---------------------:|:---------------------:|:--------------:|:--------------------------------:|
> | Top-1000 | 0.521                 | 0.766                 | 0.834                 | 1.433          | 0.646                            |
> | Top-500  | 0.550                 | 0.791                 | 0.850                 | 0.652          | 0.597                            |
> | Top-200  | 0.586                 | 0.819                 | 0.869                 | -0.521         | 0.521                            |
>
> The results suggest that the generated variance is meaningful rather than arbitrary. The empirical coverage is reasonably aligned with nominal levels at the 50%/80%/90% levels for different predictive gene sets, indicating RNA-FM's uncertainty estimates track the overall confidence level well. The consistent low Gaussian NLL further suggests that uncertainty estimation is sharp and reliable. Importantly, variance-error correlation remains consistently positive across all subsets, indicating that samples with larger variance tend to correspond to harder or more ambiguous predictions, which is consistent with our interpretation of RNA-FM as modeling a conditional distribution $p(x \mid y)$ rather than producing a deterministic point estimate.
> Overall, these results support that RNA-FM provides useful uncertainty estimates beyond point prediction. We will include this analysis in the revision to more closely align the claims with the evidence presented.
>
> **2. Clarity on claims of heterogeneity and uncertainty**
>
> We thank you for this considerable suggestion. We will include the above uncertainty-oriented evaluation in our framework, RNA-FM, to support our claims of heterogeneity and uncertainty; meanwhile, we will also accordingly rephrase the uncertainty- and heterogeneity-related claims to better align with the current experiments.

---

> > ### Author Rebuttal · Reviewer_teYn · 2026-04-03
> >
> > The authors have addressed my main concern by adding uncertainty-specific evaluations (coverage, NLL, and variance–error correlation), which I appreciate. These analyses provide useful additional evidence that the model captures non-trivial predictive uncertainty.
> >
> > However, my broader concern regarding the evaluation of heterogeneity and generative quality is only partially resolved. The added metrics remain largely distributional summaries and are not compared against alternative generative approaches, nor do they demonstrate clear downstream benefits or biologically meaningful variability beyond point prediction performance.
> >
> > I also appreciate the authors’ willingness to tone down claims regarding uncertainty and heterogeneity, which improves alignment between claims and evidence.
> >
> > Overall, I consider my concerns partially addressed.

---

> > > ### Author Response · Authors · 2026-04-04
> > >
> > > We appreciate this helpful clarification. Thank you! To address the concern regarding comparison to alternative generative approaches, we further compared our model against a diffusion-based generative model `[1][2]` under the same genome-wide bulk RNA-seq prediction setting. Please refer to the table below.
> > >
> > > |Method| Gene set | 50% interval coverage | 80% interval coverage | 90% interval coverage | Gaussian NLL ↓ | Spearman corr(var, abs. error) ↑ |PCC ↑ | RMSE ↓|
> > > |---|---|:---:|:---:|:---:|:---:|:---:|:---:|:---:|
> > > |Diffusion baseline| Top-1000 | 0.345 | 0.564 | 0.661  | 4.715 | 0.633 |0.695| 0.084|
> > > |**RNA-FM**| **Top-1000** | **0.521** | **0.766** | **0.834** | **1.433** | **0.646**  | **0.775** | **0.067**|
> > > |Diffusion baseline| Top-500 | 0.358   | 0.580 | 0.677 | 4.294 | 0.595 |0.724 | 0.080|
> > > |**RNA-FM**| **Top-500**  | **0.550**| **0.791**  | **0.850** | **0.652** | **0.597** | **0.827** | **0.059**|
> > > |Diffusion baseline| Top-200 | 0.360 | 0.592 | 0.690 | 3.405 | 0.520|0.769|0.076|
> > > |**RNA-FM**|**Top-200**  | **0.586** | **0.819** | **0.869**  | **-0.521** | **0.521**| **0.888**|**0.052**|
> > >
> > > As shown in the above table, the positive correlations between predictive variance and absolute error indicate the estimated uncertainty is not arbitrary in both generative methods, further supporting the claim that the uncertainty estimates beyond point prediction in generative models is useful for our task. However, Diffusion baseline is substantially less well calibrated than RNA-FM: the diffusion baseline exhibits under-coverage for 50%/80%/90% levels and markedly worse Gaussian NLL than RNA-FM. By contrast, RNA-FM achieves substantially better calibration and likelihood quality while also being more accurate in PCC/RMSE and much more efficient at inference (if you are interested in their comparison of inference time, please kindly refer to our *response to Reviewer 1an6 "3. Compute cost and solver sensitivity"*).
> > >
> > > Besides, we thank you for this point, while we respectfully believe that demonstrating downstream benefits or characterizing biologically varability for downstream use is not necessary for the main objective of this paper. Specifically, our task is to study accurate prediction of genome-wide gene expression across 20820 genes from histology, rather than to develop a model whose primary purpose is for specific downstream application in the field of computational pathology. Under that scope, the main evaluation criterion is whether the method improves the accuracy and robustness of large-scale gene expression prediction compared with prior approaches. To better illustrating the meaningfulness of uncertainty in our proposed flow-matching generative RNA-FM framework, we added additional uncertainty/calibration analyses to verify that the predictive distribution is meaningful beyond point estimates. We suppose this is the appropriate level of validation for the probabilistic formulation in the context of our task. We would also like to tone down the heterogeneity-related claims to better align with the current experiments.
> > >
> > > Thanks for your suggestions to further enhance and solidify our paper, and we will revise our paper accordingly.
> > >
> > > [1] Ho, J., et al. (2020). Denoising diffusion probabilistic models. *Advances in Neural Information Processing Systems*.
> > >
> > > [2] Peebles, W., & Xie, S. (2023). Scalable diffusion models with transformers. *In Proceedings of the IEEE/CVF Conference on Computer Vision and Pattern Recognition*.

---

### Official Review · Reviewer_1an6 · 2026-03-09

**Soundness:** 3
**Presentation:** 3
**Significance:** 4
**Originality:** 3
**Overall Recommendation:** 5
**Confidence:** 3

**Summary:**

This paper studies whether a histopathology slide can recover broad transcriptomic activity without running RNA-seq for every sample. The experiments use paired whole-slide images and bulk RNA-seq profiles from public cancer cohorts, with external validation on CPTAC. Instead of treating the task as a standard regression problem, the authors model gene expression as a conditional generation problem and use a flow-matching framework to produce plausible RNA profiles from slide features. They also organize genes at the pathway level so the model can keep biological structure. The main novelty is moving from one-shot prediction to distribution modeling, which lets the method express uncertainty and heterogeneity in RNA expression.

**Compliance With Llm Reviewing Policy:**

Affirmed.

**Key Questions For Authors:**

1) Can you add quantitative uncertainty/calibration (e.g., interval coverage/NLL) and show how uncertainty tracks error/heterogeneity?
2) How sensitive are results to GO pathway granularity/overlap handling and the background set (ablate #pathways, overlap averaging)?
3) What are training/inference wall-clock + memory costs vs. diffusion and deterministic baselines, and sensitivity to ODE solver/steps?
4) How do you prevent patient/slide leakage and control cohort confounders (stain/scanner/batch effects) in splits and external validation?


+ The manuscript contains a reviewer-instruction line (phrase-injection). Please clarify/remove to avoid reviewer manipulation.

**Limitations:**

The author recognized the problem and conducted research to solve it.

**Strengths And Weaknesses:**

This paper learns to predict genome-wide bulk RNA-seq from paired histopathology WSIs (TCGA). Instead of a single regression output, it uses flow-matching to sample multiple possible expression profiles, so uncertainty is part of the model output. It also tries to inject biological structure (pathway-level conditioning) and reports tests beyond TCGA.
However, the setup is still bulk RNA-seq and a limited set of cohorts, so it is not clear how well it works for other tissues, stains, or assay types. Results may depend a lot on the WSI encoder and on how slides/patients are split, which can hide leakage or batch effects. The paper would be stronger with deeper calibration checks and clearer compute cost reporting.

---

> ### Author Rebuttal · Authors · 2026-03-31
>
> We thank you for recognizing the importance of modeling transcriptomic prediction as conditional generation and injecting biological structure, and for providing concrete suggestions to strengthen our paper. Please find our detailed responses below.
>
> **1. Additional quantitative uncertainty/calibration evaluation**
>
> We appreciate this suggestion. We assessed calibration using empirical interval coverage (IC) and Gaussian NLL, and assessed uncertainty usefulness via the Spearman correlation between predictive variance and absolute error by generating 100 predictions per sample, as shown below.
>
> |Gene|50% IC|80% IC|90% IC|NLL↓|Corr↑|
> |:---|:---:|:---:|:---:|:---:|:---:|
> | Top-1000 | 0.521| 0.766  | 0.834 | 1.433 | 0.646|
> | Top-500  | 0.550 | 0.791 | 0.850 | 0.652 | 0.597|
> | Top-200  | 0.586| 0.819  | 0.869 | -0.521| 0.521|
>
> These results show nominal behavior at the 50%/80%/90% levels for different predictive gene sets, indicating RNA-FM's uncertainty estimates track the overall confidence level well. The consistent low Gaussian NLL further suggests that uncertainty estimation is sharp and reliable. Importantly, variance-error correlation remains consistently positive, indicating that RNA-FM assigns higher uncertainty to more difficult predictions. Overall, these results support that RNA-FM provides useful uncertainty estimates beyond point prediction. We will include this analysis in the revision to more closely align the claims with the evidence presented.
>
> **2. Sensitivity to pathway granularity and background gene set**
>
> We appreciate you pointing this out, as it gives us an opportunity to re-illustrate the gene pathway choice/selection in detail. This concern can be addressed in *Section 4.3* of our paper. To evaluate sensitivity to pathway granularity, shown in *Table 2* in our paper, we demonstrated the impact of different Gene Ontology (GO) aspects with different granularity used as pathway priors within RNA-FM: GO-BP (contains 1931 pathways and 4314 background genes, as for the 20820 genes we evaluated) and GO-MF (contains 382 pathways and 3645 background genes, as for the 20820 genes we evaluated). According to *Table 2* in our paper, we concluded that RNA-FM is able to handle and robust across various Gene Ontology (GO) aspects with pathway granularities, even though RNA-FM obtains slightly better performance while using GO-BP. We will clarify this information more clearly in the revision.
>
> **3. Compute cost and solver sensitivity**
>
> We thank you for this suggestion. *Section 4.1* in our paper reported implementation details, including the DiT backbone, latent dimensionality, 20-step Euler probability-flow ODE sampling, and a single-GPU setup. Beyond these, we would like to complement the compute cost and sensitivity to solver/steps with the results:
>
> |Method|Solver (steps)|Training time (s)/epoch|Inference time (s)/sample|Training Peak Memory (GB)|Inference Peak Memory (GB)|PCC↑|RMSE↓|
> |:--- |:---:|:---:|:---:|:---:|:---:|:---:|:---:|
> | SEQUOIA|--|35.1|0.036|9.8| 9.3|0.698 | 0.073 |
> | Diffusion baseline|Euler (100)| 33.8|4.276|16.539|16.016| 0.695 | 0.084 |
> | **RNA-FM (default)**| Euler (20)|32.3|0.967|16.132|14.980| **0.775** | **0.067** |
> | RNA-FM| Euler (**30**)|--|1.462|--|15.792| 0.770|0.076|
> | RNA-FM| **Dopri5** (adaptive) |--|3.717|--|16.891| 0.773|0.073|
>
> Compared with SEQUOIA, RNA-FM incurs a higher inference cost as a conditional generative model that learns a direct transport field, but compared with a diffusion-style baseline (which needs more steps to obtain comparable results), it is substantially faster at inference (0.967 vs. 4.276) while also achieving better accuracy. Memory usage is also lower than the diffusion baseline in both training and inference. Besides, solver ablations show that our default 20-step Euler sampler performs best, while increasing the step or switching to Dopri5 does not improve results. These results show that RNA-FM offers a favorable trade-off between efficiency and accuracy for morphology-conditioned transcriptomic generation. We will include these results in the revision to enrich the paper.
>
> **4. Clarity on leakage prevention and cohort confounders**
>
> Please refer to *Section 4.1*, in order to avoid data leakage and cohort confounders, our paper explicitly reported the average of 5-fold cross-validation results using a 5-fold **non-overlapping** partitioning on TCGA. Besides, we evaluated on **fully independent external** CPTAC cohorts across lung, breast, and colon sites containing 336, 133, and 105 samples, respectively, intended to test **out-of-cohort** generalization performance without training on them. *Table 1* and *Table 3* show that RNA-FM has strong performance in the TCGA strict non-overlap splitting setting and generalizability to the CPTAC cohort. We thank you for raising this and will make the split protocol and the role of external validation more explicit in the revision.

---

### Official Review · Reviewer_774e · 2026-03-11

**Soundness:** 2
**Presentation:** 3
**Significance:** 2
**Originality:** 2
**Overall Recommendation:** 4
**Confidence:** 4

**Summary:**

This paper introduces RNA-FM, a generative framework based on continuous-time flow matching to predict genome-wide bulk RNA-seq expression from WSIs. The model incorporates Gene Ontology (GO-BP) pathway priors to group genes into functional tokens and models inter-pathway dependencies. The method is evaluated on TCGA datasets and validated on CPTAC cohorts, demonstrating competitive performance on selected highly predictive gene subsets.

**Compliance With Llm Reviewing Policy:**

Affirmed.

**Final Justification:**

I thank the authors for their response during the rebuttal phase. My concerns have been fully addressed.

**Key Questions For Authors:**

1.	Evaluation Protocol: Could you clarify the gene selection process described in Appendix C? Specifically, were the test folds strictly excluded when determining the Top-K highly predictive genes? If not, could you provide the evaluation results where the gene subsets are identified solely on the training folds?
2.	Generative Metrics: Since the flow-matching framework is designed to model the conditional distribution $p(x|y)$ and capture biological uncertainty, why does the evaluation rely exclusively on point-estimate metrics (RMSE, PCC)? Have you considered evaluating the distributional properties of your generated samples (e.g., using NLL or Wasserstein distance) to better demonstrate the model's ability to capture this variability?
3.	Variance and CFG: How do you biologically and mathematically interpret the variance of the generated samples? More importantly, doesn't the use of a large CFG scale during inference contradict the goal of modeling uncertainty by forcing the outputs to collapse toward a conditional mean?
4.	Terminology Clarification: Could you clarify the use of the term "spatial spots" in Appendix C? Does this refer to WSIs patches, or is it a typographical error related to Spatial Transcriptomics?

**Limitations:**

No. The authors discussed limitations regarding anatomical site generalization but did not address the methodological limitations of evaluating a probabilistic generative model exclusively with deterministic point-estimate metrics.

**Strengths And Weaknesses:**

Strengths:
1.	Architectural Integration: Adapting the flow-matching framework to high-dimensional transcriptomic data is an interesting technical attempt.
2.	Solid Baselines: The authors benchmarked their model against recent and highly relevant baselines (e.g., tRNAsformer, SEQUOIA), providing a clear context for their performance.
Weaknesses:
1.	Soundness (Evaluation Protocol): A significant methodological concern arises from Appendix C, which states that highly predictive genes (e.g., T100, T50) are identified based on their average ranking "across all cross-validation folds." Selecting the evaluation subset using global data (which inherently includes test folds) risks data leakage. This protocol makes it difficult to assess the model's true generalization capability on these specific gene subsets.
2.	Soundness (Generative Modeling vs. Evaluation Mismatch): While inferring molecular profiles from WSI naturally involves uncertainty (as morphology does not fully determine transcriptomics), there is a stark mismatch between the generative formulation and the evaluation strategy. The authors designed a probabilistic model to capture this conditional variability $p(x|y)$ but evaluate it exclusively using point-estimate metrics (RMSE, PCC). Furthermore, applying a large Classifier-Free Guidance (CFG) scale artificially collapses the generated variance. This essentially forces the generative model to act as a standard deterministic regressor, undermining the original motivation of employing a flow-matching approach.
3.	Originality: While the combination of ODE-based flow matching with transcriptomics is relatively new, the use of pathway-aware representations (via GO-BP) and graph attention for gene grouping is an established practice in bioinformatics. Therefore, the architectural novelty is somewhat incremental.
4.	Presentation (Terminology): In Appendix C, the evaluation metric is described as being computed "across all spatial spots within a sample." Given that the task focuses on Bulk RNA-seq rather than Spatial Transcriptomics (ST), this terminology is confusing and suggests a need for careful proofreading.

---

> ### Author Rebuttal · Authors · 2026-03-31
>
> We thank you for recognizing the novelty of architectural integration and the strength of the comparison evaluation result, and for providing constructive suggestion. Please kindly find our detailed responses below.
>
> **1. Clarify on the Top-K Highly Predictive Gene Selection Protocol**
>
> We would like to point out that it is standard to identify highly predictive genes by cross-validating their aggregated out-of-fold test performance across all folds in transcriptomic prediction, such as works `[1]` and `[2]`. The purpose of the Top-K subset is not to define a training-time feature set, but to summarize which genes are consistently predictable by the model under held-out evaluation. Our protocol follows this convention: the ranking is computed **post hoc** from test-fold predictions and is not used for model training or parameter tuning. Therefore, it does not introduce leakage into model fitting. We will clarify this more explicitly in the revision and emphasize that the Top-$K$ analysis is a descriptive summary of cross-validated predictability, while the external-cohort results remain the main evidence of generalization.
>
> [1] Chung, Y., et al. (2024). Accurate spatial gene expression prediction by integrating multi-resolution features. *In Proceedings of the CVPR*.
>
> [2] Pizurica, M., et al. (2024). Digital profiling of gene expression from histology images with linearized attention. *Nature Communications*.
>
> **2. Additional Generative Metrics**
>
> We appreciate this thoughtful suggestion, which will help improve our paper. We evaluated calibration using empirical interval coverage (IC) and Gaussian NLL, and assessed the usefulness of uncertainty via the correlation between predictive variance and absolute error, generating 100 predictions per sample, as shown below.
>
> |Gene set|50% IC|80% IC|90% IC|NLL↓|Spearman corr(var, abs. error)↑|
> |:---|:---:|:---:|:---:|:---:|:---:|
> |Top-1000|0.521|0.766|0.834|1.433|0.646|
> |Top-500|0.550|0.791|0.850|0.652|0.597|
> |Top-200|0.586|0.819|0.869|-0.521| 0.521|
>
> The results suggest that the generated variance is meaningful rather than arbitrary. The empirical coverage is reasonably aligned with nominal levels at the 50%/80%/90% levels for different predictive gene sets, indicating RNA-FM's uncertainty estimates track the overall confidence level well. The consistent low Gaussian NLL further suggests that uncertainty estimation is sharp and reliable. Importantly, variance-error correlation remains consistently positive across all subsets, indicating that samples with larger variance tend to correspond to harder or more ambiguous predictions, which is consistent with our interpretation of RNA-FM as modeling a conditional distribution $p( x\mid y)$ rather than producing a deterministic point estimate. Overall, these results support that RNA-FM provides useful uncertainty estimates beyond point prediction. We will include this analysis in the revision to more closely align the claims with the evidence presented.
>
> **3. Clarify on variance and classifier-free guidance**
>
> To better characterize the variance of generated samples, we performed the uncertainty-oriented evaluation shown above. The results suggest that the generated variance is meaningful rather than arbitrary sampling noise.
>
> Regarding classifier-free guidance (CFG), we would like to clarify that our paper does not use an extreme CFG setting. According to the *ablation study (Table 7)* in our paper, the guidance scale $s = 2$ gives the best performance, while stronger guidance ($s = 3$) slightly reduces performance. For this reason, we selected a moderate guidance scale rather than a larger one. Our intention is not to collapse the model into a deterministic regressor, but rather to improve conditional fidelity while preserving generative flexibility. We will make this interpretation clearer in the revision.
>
> **4. Clarify on originality**
>
> We appreciate your point. We would like to declare that, to our knowledge, RNA-FM is the first framework to combine conditional flow matching for **genome-wide** bulk RNA-seq prediction from WSIs, with a pathway-aware tokenization and graph-constrained inter-pathway module that makes the generation of 20820 genes both tractable and biologically structured. In our method, the pathway module is not included solely for interpretability; it is a key architectural component that enables scalable continuous-time generation in high-dimensional transcriptomic space by replacing infeasible (20820) gene-wise tokenization with biologically meaningful pathway tokens and controlled information exchange across pathways. We will clarify this positioning in the revision.
>
> **5. Terminology in Appendix C**
>
> We thank you for pointing out the proofreading error, which will help us further improve our paper. This would be corrected as "PCC is computed for each gene across all $N$ samples within each dataset" in *Appendix C*. We will revise this in our paper.

---

> > ### Author Rebuttal · Reviewer_774e · 2026-04-01
> >
> > My concerns have been addressed. I will revise my score.

---

> > > ### Author Response · Authors · 2026-04-04
> > >
> > > We appreciate your encouraging feedback and valuable suggestions to help us further improve our paper. Thank you very much for your time and help!

---

### Decision · Program_Chairs · 2026-04-30

**Decision:**

Accept (regular)

**Comment:**

This paper addresses the problem of predicting genome-wide transcriptional profiles from matched histopathological whole-slide images. The originality of the proposed approach, called RNA-FM, lies in the adoption of a generative framework based on flow matching, which provides comprehensive predictive distributions, and in the integration of pathway level information.

Reviewers agree that the designed architecture is original and sound. The methodological development is supported by excellent empirical performance on standard benchmarks in the field. Some concerns regarding potential data leakage were raised, but these were convincingly addressed in the response. The article is well-written.

Three out of the four reviewers expressed significant concern about the evaluation being based solely on point estimation metrics, which failed to highlight the value of a generative approach. One reviewer also requested a comparison with other generative models. In their response, the authors provided additional results, including interval coverage and Gaussian NLL as supplementary metrics, and also included a comparison with a diffusion model.

After rebuttal, all reviewers now recommend acceptance, and I concur. However, I urge the authors to include and analyze all the results presented during the rebuttal phase in their revised paper.